# The Impact Relationships between Scientific and Technological Innovation, Industrial Structure Advancement and Carbon Footprints in China Based on the PVAR Model

**DOI:** 10.3390/ijerph19159513

**Published:** 2022-08-03

**Authors:** Shengli Dai, Yingying Wang, Weimin Zhang

**Affiliations:** 1School of Public Administration, Central China Normal University, Wuhan 430079, China; storymaker@163.com (S.D.); zhangweimin@mails.ccnu.edu.cn (W.Z.); 2School of Public Administration and Emergency Management, Jinan University, Guangzhou 510632, China

**Keywords:** Beijing–Tianjin–Hebei urban agglomeration, scientific and technological innovation, industrial structure advancement, carbon footprints, panel vector auto-regressive model

## Abstract

As one of the three major engines of economic growth in China, the Beijing–Tianjin–Hebei (BTH) urban agglomeration has become one of the regions with the highest energy consumption intensity. To investigate the dynamic relationships between scientific and technological innovation, industrial structure advancement and carbon footprints, panel data in BTH from 2006 to 2019 was selected, and a Panel Vector Auto-Regressive (PVAR) model was established to conduct an empirical study. The conclusions show that there is a causal relationship between the industrial structure advancement and carbon footprints, and the influence of each on the other is significant. The impact of scientific and technological innovation on carbon footprints has a “rebound effect”. Scientific and technological innovation can accelerate the process of industrial structure advancement. Carbon footprints have a significant backward forcing effect on both industrial structure advancement and scientific and technological innovation, with impact coefficients of 0.0671 and 0.2120, respectively. Compared with scientific and technological innovation, the industrial structure advancement has a greater impact on carbon footprints, with a variance contribution of 25.4%. The research findings are conducive to providing policy support for the coordinated development of BTH and promoting the realization of the Double Carbon goal.

## 1. Introduction

Since 2020, the global climate governance system has entered a new phase with the Paris Agreement as its core. Climate change has become a global public issue due to the massive emission of CO_2_ and other greenhouse gases. The international community has been paying more attention to carbon neutrality and other concepts. The climate governance actions have been intensifying, but the global efforts on climate governance have not shown obvious results [1]. According to *Global Energy Review: CO_2_ Emissions in 2021* published by the International Energy Agency (IEA), global CO_2_ footprints from the energy sector reached 36.3 billion tons in 2021, up more than 2 billion tons from 2020 and up 6% year-over-year [2]. The record growth has offset the decline in carbon footprints brought about by weaker economic activity since the Newcastle pneumonia epidemic. The global economy is still highly dependent on fossil fuels, as shown by national economic growth data. Global GDP will rise by about 5.9% year-on-year in 2021, essentially the same as the rise in carbon footprints, which means that the global economic recovery is closely linked to carbon footprints [3]. Therefore, on the premise of ensuring energy security, the use of green technology innovation to accelerate the transformation of industrial and energy structures, so as to achieve a reasonable control of global carbon footprints is now, more than ever, necessary.

With China’s rapid economic development and expansion of production activities, carbon footprints are on the rise. Currently, China surpasses the United States as the world’s top carbon emitter. According to the International Energy Agency (IEA), China’s overall carbon footprints have nearly doubled from 2005 to 2019 [2]. Measures to reduce carbon footprints are critical to long-term economic prosperity. In July 2008, eight countries at the G8 summit signed the *United Nations Framework Convention on Climate Change*, together agreeing to a long-term goal of reducing global greenhouse gas emissions by 50 percent by 2050. In September 2020, China announced that China strives to peak its CO_2_ emissions by 2030 and works to achieve its carbon neutrality goal by 2060 at the 75th session of the United Nations General Assembly [4]. Technological innovation and industrial restructuring are important ways for China to meet its carbon reduction targets and maintain sustainable economic development. At the same time, the emission reduction targets are also forcing governments and enterprises to undertake technological innovation and transform industrial structure.

With the capital at its core and a reform-led zone for collaboration, the Beijing–Tianjin–Hebei (BTH) urban agglomeration is one of the three major engines of China’s economic growth as a world-class city cluster. In April 2015, the *Beijin**g**–**Tianjin**–**Hebei Synergistic Development Planning Outline* proposed that the synergistic development of BTH is a major national strategy to vigorously promote innovation-driven development and to take the lead in making breakthroughs in ecological environmental protection and industrial upgrading. However, as one of the larger industrial bases in China, the concentration of high-energy-consuming and high-polluting industries has made BTH one of the regions with the highest energy consumption intensity and the most serious air pollution. The single coal-based energy consumption structure poses a great threat to the development of a low-carbon economy [5,6]. At present, carbon footprints from the BTH are huge and on the rise, and the efficiency of carbon footprints also needs to be improved [7]. Moreover, there are obvious spatial differences in carbon footprints’ efficiency, both at the regional development level and coordination level [8,9]. As a demonstration area for the collaborative construction of regional ecological civilization, how to enhance industrial energy utilization efficiency and reduce pollution emissions to synergistically improve the ecological environment has become a major issue to be solved. Therefore, we need to clarify the relationships between scientific and technological innovation, industrial restructuring and carbon footprints in BTH, and explore the intrinsic influence mechanism. This has great significance for achieving the 2030 emission reduction target and promoting high-quality economic development.

In this study, we aimed to investigate the dynamic relationships between scientific and technological innovation, industrial structure advancement and carbon footprints in the BTH region. Since Beijing and Tianjin are municipalities directly under the central government, and Hebei province is a large region with prefecture-level cities of different development levels from north to south, we selected prefecture-level cities and municipalities directly under the central government in the BTH as the unit of analysis. This also facilitates a detailed analysis of the actual situation of each city. Specifically, this study aims to address the following questions. First, whether there are interaction relationships between scientific and technological innovation, industrial structure advancement and carbon footprints. Second, whether there are causal relationships between scientific and technological innovation, industrial structure advancement and carbon footprints. Third, what is the process of mutual influence among scientific and technological innovation, industrial structure advancement and carbon footprints over time?

In order to answer the above questions, we first numerically calculated the indexes of scientific and technological innovation, industrial structure advancement and carbon footprints, and analyzed the spatial and temporal evolution of carbon emissions in the BTH. After passing the smoothing test, we selected the optimal lag order for Generalized Method of Moments estimation to derive the static influence relationships between the three variables. Then, Granger causality was used to test whether there are causal relationships. The direction of interaction between the variables and their dynamic relationships in different periods was explored in depth by impulse response function. Finally, based on the research results, policy recommendations are proposed for the development status of the BTH.

The rest of this study is organized as follows. In Section 2, we briefly review the relevant work in the previous literature. Section 3 presents the data sources, variable descriptions, and methodology. In Section 4, the spatial and temporal evolution is analyzed. In Section 5, the empirical analysis of this study is presented. In Section 6, we summarize the conclusions and discuss the limitations. In Section 7, we show the research implications based on the results and findings.

## 2. Literature Review

At present, scholars have widely discussed carbon footprints and its influencing factors from different perspectives. Among them, the literature on the relationships between scientific and technological innovation, industrial restructuring and carbon footprints mainly concerns in the following three aspects.

### 2.1. Scientific and Technological Innovation and Industrial Restructuring

There are two main views on the study of the relationship between scientific and technological innovation and industrial structuring. One is that there is a “threshold effect” between them. Scholars have carried out empirical analysis from different dimensions based on different research scopes. In terms of direct impact, technological innovation can effectively promote the rationalization of industrial structure, and there is a “U-shaped” curve relationship with the industrial structure advancement [10,11]. In terms of specific paths of action, technological and institutional innovation drive can promote industrial structure upgrading by strengthening the concentration of high-tech industries, stimulating consumption demand in markets, and promoting the level of population aging [12,13]. Other scholars have characterized industrial structure upgrading from the structural indexes of different dimensions, and empirically tested the impact of green technological innovation and the moderating effect therein, indicating that green technological innovation can effectively promote industrial structure upgrading. There is a “threshold effect” on the impact of green technological innovation on industrial structure upgrading. Moreover, the impact of green technology innovation on industrial structure upgrading is different under the degree of market distortion of different factors [14,15].

Second, there is an “intermediary effect” between industrial structure and scientific and technological innovation. From the perspective of industrial structure rationalization and industrial structure advancement, scholars have found that both industrial structure rationalization and advanced industrial structure have a significant role in promoting the relationship between green technology innovation in industrial enterprises and high-quality economic development [16,17]. From the perspective of scientific and technological innovation heterogeneity, scholars divided innovation into technological innovation and product innovation, and found that technological innovation and product innovation promote economic growth through industrial structure upgrading via the mediating effect model [18]. In addition, other scholars, by analyzing the experience of science and technology innovation to promote industrial structure upgrading, found that government policy intervention has limitations, which can stimulate the rapid development of supported industries in the short term, but at the same time, it can also trigger the problem of homogeneous investment and construction overheating, leading to technological convergence [19,20].

### 2.2. Scientific and Technological Innovation and Carbon Footprints

As for the relationship between scientific and technological innovation and carbon footprints, there are two main views. One view is that scientific and technological innovation directly inhibits the growth of carbon footprints. Scholars have examined the influence mechanism between them by applying the spatial econometric model, fixed effect model and random effect model. The results show that different types of green innovation technologies have a significant inhibitory effect on carbon footprints [21,22], and the direct impact of breakthrough low-carbon technology innovation on carbon footprints with “spatial spillover effects”, both in the short and long term, presents a significant inhibitory effect on carbon footprints [23]. For the relationship between carbon footprints intensity, total carbon footprints and technological innovation, technological innovation efficiency has a significant negative effect on both carbon footprints intensity and total carbon footprints [24]. An in-depth analysis of technological progress and scale efficiency revealed that technological progress in carbon footprints reduction and scale efficiency played a positive role in carbon footprints, but the effect of technological progress in energy efficiency was lower than that of increasing the scale efficiency of technological progress in energy [25].

Another view is that scientific and technological innovation indirectly affects carbon footprints. The reason is that low-carbon technological innovation indirectly suppresses the increase in carbon footprints by changing the energy structure, which has a negative effect on low-carbon technological innovation, and energy intensity is the main inhibitor of carbon footprints growth in industrial economies [26]. The Khazzoom−Brookes hypothesis also points out the “rebound effect” of energy. Technological innovation promotes economic growth by increasing energy efficiency [27], but economic growth increases the demand for energy [28]. The “rebound effect” can lead to a significant increase in output level and energy consumption together, and thus does not determine the relationship between technological innovation and carbon efficiency [29]. Other scholars believe that industrial restructuring guided by technological innovation is the real reason for reducing energy consumption and having a positive effect on carbon footprints reduction [30]. Many scholars also point out that there is spatial heterogeneity in the impact of technological innovation on carbon emissions. Compared with technological innovation outputs, technological innovation inputs are more effective in carbon emission reduction. In addition, there is significant heterogeneity in the carbon emission reduction effects of technological innovation in different regions. The technological innovation improvement in eastern China can reduce emission. There is a Pareto improvement in the impact of technological innovation on carbon emission reduction in the central region, while the driving effect in the western region is not significant [31].

### 2.3. Industrial Restructuring and Carbon Footprints

Many scholars have explored the role of industrial restructuring and carbon footprints. One view is that there are “intermediary effects” and “spatial spillover effects” between carbon footprints and industrial restructuring. Industrial structuring has both a direct carbon footprints reduction effect and a intermediary effect of reducing carbon footprints through technological innovation [32], and the intermediary effect of technological innovation in reducing carbon footprints remains significant in the sub-region [33,34]. The industrial structure advancement can reduce both the carbon footprints intensity of the region and even the surrounding areas [35,36]. The ability of industrial structure advancement to reduce carbon footprints intensity is the greatest compared to economic development, technological innovation and urbanization [37]. Meanwhile, industrial resource allocation efficiency and industrial structure advancement also have a significant mitigating effect on carbon footprints [38].

Another view is that there is a “crowding-out effect” between carbon footprints and industrial restructuring. Industrial structure upgrading can reduce carbon footprints in the province, and may affect the carbon footprints intensity of neighboring provinces through the crowding out effect on polluting industries [39]. As the level of economic development increases, the industrial structure rationalization can generally suppress carbon footprints, while the industrial structure advancement is characterized by an obvious “inverted U-shaped” trend, which has two sides, with developed regions showing a suppression effect and less developed regions showing the opposite [40]. In addition, scholars have studied the practices of developed countries to reduce carbon emission intensity by adjusting industrial structure. In view of the realities of China’s industrial structure, the industrialization process has not yet been completed and the high energy-consuming industries account for a large proportion [41]. Thus, it is recommended to actively cultivate new energy and other emerging industries to ensure the safety of the industrial chain and of the supply chain to promote the optimization of the industrial space layout and to promote the formation of a new development pattern [42,43].

To sum up, scholars have extensively analyzed the influence relationships among scientific and technological innovation, industrial structure advancement and carbon footprints. Related studies are shown in Figure 1, but there are still some shortcomings. First, many studies have thoroughly explored the two-two relationship and influence mechanism among carbon footprints, scientific and technological innovation and industrial structure, but there are fewer studies on the dynamic influence relationship among the three under the same theoretical framework. Second, some studies use time series data or panel regression models, ignoring the endogenous variables. Therefore, we have taken BTH as the research scope, and analyzed the dynamic relationships between scientific and technological innovation, industrial restructuring and carbon footprints by using the Panel Vector Auto-regression (PVAR) Model to provide some ideas for cracking these environmental problems.

## 3. Methodology and Data

In this section, we elaborate on the data sources and identify the variables so as to construct a PVAR model to explore whether there are dynamic correlations between carbon footprint, scientific and technological innovation and industrial structure advancement in BTH, while also laying the foundation for the subsequent empirical analysis.

### 3.1. Data Sources

The research object is the panel data of carbon footprints, industrial structure advancement and scientific and technological innovation of 13 cities in BTH from 2006 to 2019. All data are obtained from the *China Energy Statistical Yearbook*, *China City Statistical Yearbook*, *Beijing Statistical Yearbook*, *Tianjin Statistical Yearbook*, *Hebei Economic Yearbook* and statistical yearbooks of prefecture-level cities, and relevant data references are taken from the *IPCC Guidelines for National Greenhouse Gas Inventories*. The specific geographical distribution of the BTH in China is shown in Figure 2. The missing values of individual indicators are supplemented by linear interpolation.

### 3.2. Variable Descriptions

In this study, three variables are involved: carbon footprints, industrial structure advancement and scientific and technological innovation.

Carbon footprints (CF): according to the United Nations Intergovernmental Panel on Climate Change (IPCC) report, the combustion of fossil energy is the main cause of greenhouse gas emissions [44]. In addition, urban carbon footprints include the consumption of electrical and thermal energy. Firstly, we adopted the approach of Wu [45] and apply the IPCC inventory factor method to calculate the carbon footprints of each city indirectly through the consumption of fossil energy. The core work is then to determine specific emission factors for various energy consumption processes [46]. The *2006 IPCC Guidelines for National Greenhouse Gas Inventories* published by the IPCC are the most widely used by scholars, which provide rationalized recommended values for each carbon footprints factor according to the world average. Therefore, the IPCC inventory factor method was used to calculate carbon footprints from direct energy consumption in BTH. Secondly, electricity consumption was borrowed from Glaeser [47], and each regional grid was considered as an emission factor for calculation. The carbon footprints from each city’s electric energy consumption were calculated based on the baseline emission factors and urban electric energy consumption of the six regional grids published by China Power Grid. Thirdly, for the energy consumed by urban transportation, the approach of Li et al. [48] was borrowed. Assuming that the energy consumption intensity and carbon footprints intensity of each type of transportation mode are proportional, the energy consumption per unit of passenger volume (ton-kilometers) and freight volume (10,000 person-kilometers) was calculated using the various types of energy consumed in the transportation sector from the *China Statistical Yearbook*. The transportation energy consumption and carbon footprints of each city were calculated based on the passenger and freight volumes. Finally, urban heat is mainly supplied by boiler houses and thermal power plants, and its raw materials are mainly raw coal. The *China Urban Construction Statistical Yearbook* provides statistics on heat supply in each city in previous years. The minimum standard of thermal efficiency of coal-fired industrial boilers stipulated in *GB/T15317-2009 Energy Conservation Monitoring of Coal-fired Industrial Boilers* is between 65% and 78%, and a thermal efficiency value of 70% was used for calculation. The average low level heat of raw coal is 20,908 kJ/kg. The amount of raw coal required was calculated using the heat supply, thermal efficiency and raw coal heat generation coefficient. Then the amount of energy consumed for central heating was calculated using the raw coal conversion factor (0. 7143 kg of standard coal per kg). According to IPCC 2006, the carbon footprints factor is 2.5 kg CO_2_/kg per kg of raw coal; using the amount of raw coal consumed for thermal energy, the carbon footprints genrated by centralized heating can be calculated. The carbon footprints from electricity, gas and liquefied petroleum gas, transportation and thermal energy consumption are added together to obtain the total carbon footprints of each city.

Industrial structure advancement (ISA): this is manifested as the gradual replacement of factor capital-dependent low-end industries by an advanced structure dominated by knowledge and technology-intensive industries. Traditional industries begin to transform and upgrade, and this reflects the process of industrial restructuring from primary industry-based to secondary industry to tertiary industry-based. In this study, we adopt the approach of Fu [49], which divides GDP into three parts according to the three industrial divisions, and constitutes a set of three-dimensional vectors X_0_ = (X_1__,0_, X_2__,0_, X_3__,0_). Then the angles θ_1_, θ_2_, θ_3_, were calculated separately between X_0_ and the reference vectors X_1_ = (1, 0, 0), X_2_ = (0, 1, 0), X_3_ = (0, 0, 1), which are arranged from the lower to the higher levels of industries.
(1)θj=arccos[∑i=13(xi,j·xi,0)(∑i=13(xi,j2)12·∑i=13(xi,02)12)]
j = 1, 2, 3. In Equation (1), X_i,j_ is the i-th component of the basic unit vector group Xi (i = 1, 2, 3). X_i,0_ is the i-th component of vector X_0_. The computed index was then taken into Equation (2) to obtain the index of industrial structure advancement.
(2)$$\mathrm{ISA}=\sum_{k=1}^{3}\sum_{j=1}^{k}\theta_{j}$$

In Equation (2), k = 1, 2, 3. $\theta_{j}$ is the angle between the three-dimensional vector X_0_ and the reference vectors. The larger the ISA, the higher the level of industrial structure advancement.

Scientific and technological innovation (INN): the number of granted invention patents reflects the scientific and technological innovation capability and the level of innovation output of a city. Therefore, we adopted the number of granted invention patents to indicate the level of scientific and technological innovation of the city. To reduce the influence of heteroscedasticity of different data, the values were taken as logarithms. The descriptive statistics of all variables are shown in Table 1. The statistical characteristics of each variable are described in Table 2.

### 3.3. Panel Vector Auto-Regressive Model

In this study, we investigated the dynamic relationships between scientific and technological innovation, industrial structure advancement and carbon footprints, and chose the panel vector auto-regressive (PVAR) model and Gaussian Mixture Model (GMM) estimation to deal with the possible correlations and endogeneity of the variable series. The PVAR model was first proposed by Holtz-Eakin et al. [50] and then gradually refined by Mccoskey and Kao [51]. The model follows the advantages of the vector auto-regressive (VAR) model. The PVAR model treats scientific and technological innovation, industrial structure advancement and carbon footprints as endogenous variables, and analyzes the effects of each variable and its lagged variables on other variables in the model. Therefore, it combines the characteristics of large cross-section and short time series, which can overcome the restrictive conditions of the model in terms of time series and panel to some extent, and it can better reflect the influence of individual differences on the model. Based on the individual difference capture feature of the PVAR model, the model is constructed as follows.
(3)$$Y_{\mathrm{it}}=\alpha_{0}+\overset{n}{\underset{j=1}{\sum }}\alpha_{j}Y_{\mathrm{it}-j}+\beta_{i}+\gamma_{i}+\varepsilon_{\mathrm{it}}$$

In Equation (3), $\alpha_{0}$ is the intercept term; j is the lag order; i represents each city in BTH; t represents the year;${\;\alpha }_{j}$ is a column vector of order 1 × 3 containing the three endogenous variables CF, ISA, and INN, that is $Y_{\mathrm{it}}=\left[\begin{array}{c} {\mathrm{CF}}_{\mathrm{it}}\\ {\mathrm{ISA}}_{\mathrm{it}}\\ {\mathrm{INN}}_{\mathrm{it}} \end{array}\right]$; $\beta_{i}$ is the individual fixed effect; $\beta_{i\;}$ is the individual time-point effect; and $\gamma_{i}\;$ is the random perturbation term.

## 4. Evolution Analysis

### 4.1. Spatial and Temporal Evolution of Carbon Footprints

The carbon footprints in BTH show certain evolutionary characteristics in time and space. For time, we selected the carbon footprints of each city in 2006, 2010, 2015 and 2019 for analysis. The spatial distribution of carbon footprints are shown in Figure 3. Among them, the cities with large carbon footprints in 2006 wre Beijing and Tianjin. Cities in Hebei Province have lower carbon footprints. From 2006 to 2010, with the rapid development of industrial society, the consumption of large amounts of energy made the cities’ carbon emissions increase significantly. Carbon footprints in Beijing and Tianjin are still increasing significantly. In addition to cities with relatively large initial values of carbon footprints, Tangshan also belongs to the high growth cities. By 2015, people gradually realized that economic development cannot occur at the expense of the environment. Therefore, most cities had insignificant increases in their carbon footprints. In 2019, Zhangjiakou, Chengde, Qinhuangdao, Langfang, cangzhou, Xingtai and Hengshui became cities with relatively low carbon emissions, while Tangshan, Baoding, Shijiazhuang and Handan, on the other hand, had a significant increase in carbon emissions.

Spatially, the cities with the highest carbon emissions are concentrated in the central region. The carbon footprints of Beijing, Tianjin and Tangshan are much higher than those of other cities. As the political and economic center of China, the growth of the total economic volume per capita in Beijing is one of the main reasons for the high growth of its carbon footprints. The rapid development of Tianjin’s industrial economy has greatly increased its energy consumption, and industry has become the mainstay of its energy consumption, ultimately leading to an increase in carbon footprints as well. Tangshan is an important energy and raw material base in BTH, with a high proportion of heavy chemical industries. The high-carbon structure of the industrial system causes Tangshan’s energy consumption structure to be dominated by coal. The single energy consumption structure has resulted in high carbon footprints. With the deepening of urbanization, the carbon footprints of Shijiazhuang, Baoding and Handan also have different degrees of growth. In addition, Qinhuangdao, Chengde, Zhangjiakou, Cangzhou, Hengshui, Xingtai and Langfang are cities with relatively low carbon emissions. This is due to the pattern of urban development and industrial distribution.

### 4.2. Temporal Evolution of Industrial Structure Advancement

The trend of industrial structure change reflects the adjustment of the production sector and national economy structure. The industrial structure advancement of BTH from 2006 to 2019 is shown in Figure 4. It is obvious that Beijing and Tianjin have better industrial structure high polarization than the cities in Hebei province. On the whole, the level of industrial structure advancement of each city keeps improving, which indicates that the city cluster keeps allocating resources effectively and the macroscopic industrial development keeps coordinating. Among them, Langfang has the largest increase, followed by Hengshui. In 2019, the cities with a higher level of industrial structure advancement were Beijing, Tianjin, Langfang and Shijiazhuangin, with the industrial structure advancement index exceeding 7.0. This indicates that these cities are actively transforming the industrial structure from lower to higher forms and entering the industrialization stage with high manufacturing growth. However, the advanced level of industrial structure in Chengde is still low, located below 6.5. It indicates that Chengde’s industrial structure still needs to be further transformed to a high level.

### 4.3. Spatial and Temporal Evolution of Scientific and Technological Innovation

In order to explore the evolutionary trend of scientific and technological innovation in BTH, the patent grant volume of each city in 2006, 2010, 2015 and 2019 are selected for comparative analysis. Spatial distribution of patent grants and growth rates are shown in Figure 5. Obviously, there has been a significant increase in the patent grant volume of each city. The patent grant volume of some cities in 2019 even exceeded the sum of 2006, 2010 and 2015. Among them, Beijing, Tianjin and Shijiazhuang are the cities with the largest patent grant volume. In 2019, Beijing’s patent grant volume exceeded 130,000 pieces. In terms of growth rates, Zhangjiakou, Langfang and Xingtai have seen breakthrough growth in patent grants in recent years, using 2006 as a benchmark. This reflects that city developers are paying more and more attention to science and technology innovation, and the related mechanism of certifying and transforming achievements has become better.

## 5. Empirical Analysis

### 5.1. Smoothing Test and Optimal Lag Order Determination

#### 5.1.1. Smoothing Test

Before estimating the PVAR model, it is necessary to conduct a smoothing test for each panel series to avoid bias in the results due to pseudo-regression. Therefore, in order to avoid the test error caused by the single method test and ensure accuracy, we adopt four methods: LLC test, IPS test, Fisher−ADF and Fisher−PP test to examine the unit root of each variable at the same time. The results of the panel unit root test for each variable are shown in Table 3. The results show that after first-order differential, the original hypothesis of the existence of unit root in the panel data was rejected at least at the 5% significance level. Therefore, it passed the smoothing test.

#### 5.1.2. Co-Integration Test

After the smoothing test, we have used Kao [52] and Pedroni [53] to conduct co-integration tests for the three variables of CF, ISA and INN, and the test results are shown in Table 4. The original hypothesis was rejected at the 1% level for all variables, indicating the existence of a long-run co-integration relationship between the variables. Therefore, the PVAR model could be constructed.

#### 5.1.3. Optimal Lag Order

To construct the PVAR model, we determined the optimal lag order of the model, i.e., the order where the minimum value of the statistic is located, according to the Akuchi Information Criterion (AIC), Bayesian Information Criterion (BIC), and Hannan−Quinn Information Criterion (HQIC) [54]. Optimal lag order determination results are shown in Table 5. The lag order chosen should not be too large, otherwise it will reduce the degrees of freedom of the model and cause unnecessary loss of model data. A too-small lag order will reduce the accuracy of the model test results. Therefore, the selection should be based on the principle of passing a larger number of test criteria [55]. The results show that the smallest value of the model statistic is of lag order 1 for three criteria. Therefore, in this PVAR model, the optimal lag order is chosen to be lag order 1.

### 5.2. GMM Estimation Based on the PVAR Model

By building a PVAR model with a lag order 1, we applied a Generalized Moment Model (GMM) for estimation. To eliminate endogeneity in the PVAR model, we used the Mean Difference Method and Helmert Method to remove time point effects and fixed effects. h_CF, h_ISA, and h_INN were the variables after performing the helmert transformation, and L1 denotes lag order 1. The estimation results are shown in Table 6.

As can be seen from Table 5, when the h_CF equation is used as the dependent variable, h_CF with one period lag has a significant positive contribution to itself at the 1% level, with an impact coefficient of 0.9987. h_ISA with one period lag has a significant negative impact on h_CF at the 5% level, with an impact coefficient of −0.8766. It shows that industrial structure advancement has a more obvious inhibitory effect on carbon footprints on the basis of the industrial structure rationalization. The coefficient of h_INN has a significant positive effect on h_CF at the 10% level, with a lag of 0.0992, indicating that there is a “rebound effect” on the impact of technological innovation on carbon footprints. On one hand, lower production costs will lead to greater external demand for production, which in turn will increase carbon footprints. On the other hand, with the narrowing of the technology emission reduction space and the gradual acceleration of the carbon market construction, the cost of carbon footprint reduction is bound to continue to rise. Thus, the single measure of scientific and technological innovation will lead to an increase in carbon footprints in the long run.

When the h_ISA equation is used as the dependent variable, h_CF with one period lag has a significant positive effect on h_ISA at the 1% level with an impact coefficient of 0.0671, indicating that the increase in carbon footprints in the short term will force the government and enterprises to upgrade the industrial structure. h_ISA with one period lag has a significant self-promoting effect on itself at the 1% level, with an impact coefficient of 0.5334. h_INN with one period lag has a significant positive impact on h_ISA at the 1% level, with an impact coefficient of 0.0282. The above results indicate that scientific and technological innovation accelerates the transformation of industries from low-end to high-tech, from factor-intensive to knowledge-intensive and from high pollution to green and low-carbon. Therefore, its positive contribution to the industrial structure advancement gradually appears.

When the h_INN equation is used as the dependent variable, h_CF in the lagged period has a significant positive effect on h_INN at the 5% level, indicating that the long-term growth of carbon footprints will force cities to improve the level of scientific and technological innovation, accelerate the innovation-driven transformation and apply advanced low-carbon scientific and technological innovation results to achieve carbon footprints reduction targets. The current h_ISA will have a significant negative inhibitory effect on h_INN at the 5% level, with an impact coefficient of −1.0002. The reason is that there is a negative mediating effect in the process, which inhibits the output of scientific and technological innovation in the short term. The h_INN in the current period will have a significant positive effect on itself at the 1% level, with an impact coefficient of 0.9089, indicating that scientific and technological innovation has some self-reinforcing effect.

### 5.3. Granger Causality Test

The above GMM estimation results indicate the static relationship between CF, ISA and INN. In order to further determine the causal relationships between three variables, the Granger causality test was conducted to verify. The results are shown in Table 7.

From above, it can be seen that CF and INN are Granger causes, CF and ISA are Granger causes, and ISA and INN are Granger causes. The dynamic relationships between the three can be seen by combining Table 5 and Table 6.

Scientific and technological innovation and carbon footprints show a two-way causal relationship. For one, the level of scientific and technological innovation affects carbon footprints. In the case of BTH, it leads to a “rebound effect”. While the efficiency of energy use is improved by scientific and technological innovation, it also increases the demand for energy in production. In addition, technological innovation will indirectly affect carbon footprints by promoting industrial restructuring. For another, the change in carbon footprints will also affect scientific and technological innovation in turn. The continuously high carbon footprints also makes enterprises and technology sectors accelerate to provide practical methods in the three aspects of low carbon, zero carbon and negative carbon to achieve technological breakthroughs and change the current carbon footprints in the long run.

The industrial structure advancement and carbon footprints show a two-way causal relationship. The industrial structure advancement is the Granger cause of carbon footprints, and the realization of the carbon footprints reduction target depends on the green transformation of industry. The green industrial transformation and upgrading in the process of industrial structure advancement promotes its transformation from factor-driven to innovation-driven by enterprises with the support of innovation achievements, which helps to realize the two-way goal of energy saving and carbon reduction. Carbon footprints are the Granger cause of the industrial structure’s advancement. Carbon footprints have a positive promotion effect on the industrial structure’s advancement. The increasing carbon footprints makes the industry change from the production mode of high energy consumption, high pollution and high emission to the green production mode. Meanwhile, the speed of inter-industry linkage and integration is accelerated to enhance resource allocation efficiency and industrial coupling.

Scientific and technological innovation and the industrial structure advancement also show a two-way cause-and-effect relationship. Scientific and technological innovation is the Granger cause of the industrial structure advancement. Green low-carbon scientific and technological innovation achievements can effectively increase the supply of low-carbon products and innovate green services and guide green consumption. By strengthening basic industrial research and improving the industrial technology innovation system can effectively increase the supply of environmental protection equipment and low-carbon products. Low-carbon products will, in turn, promote the green and low-carbon transformation of traditional industries such as transportation and construction and the rapid development of new and high-tech industries, thus promoting the high polarization of industrial structure and ultimately realizing green services and green consumption [56]. Industrial structure advancement is the Granger cause of the scientific and technological innovation. In the long term, based on the industrial structure, the deep integration of the innovation chain and industrial chain can accelerate all kinds of technology and innovation elements to consolidate green innovation achievements, promoting faster output of scientific and technological innovation.

### 5.4. Impulse-Response Analysis Based on the PVAR Model

The impulse response function (IRF) can further describe the direction of interaction between variables and their dynamic relationships at different period lags. Therefore, we conducted Impulse-response analysis for CF, ISA and INN, and give each variable a shock of 1 standard deviation, setting the lag period to 10 periods. The impulse-response function results are shown in Figure 6. After 200 Monte -Carlo simulations, the impulse responses corresponding to each variable were plotted. Errors are 5% on each side generated by Monte -Carlo with 200 reps. The Impulse-response plots were drawn after 200 Monte -Carlo simulations. The middle line is the IRF curve, and the outer lines represent the 5% and 95% quantile lines. The response trends of CF, ISA and INN converge to 0 after 10 periods, which indicates that the PVAR model is robust.

Based on Figure 6, the following conclusions can be drawn: 

When CF is subjected to a shock of one standard deviation, the impulse responses to itself, ISA and INN are as follows. Among them, CF shows a significant strong positive response after being shocked by itself, which gradually weakens until convergence with the extension of the response period, indicating the relative economic inertia of carbon footprints in the BTH. After the shock to CF, ISA indicates a significant positive effect, which reaches a peak in the third period and then gradually decreases until it converges to 0. The positive effect of CF on INN gradually increases in the first period, and the response degree starts to weaken after the third period, and then gradually converges to 0. It can be seen that the impact of carbon footprints in BTH has a long-term nature and has an obvious push-back effect on the industrial structure advancement and technological innovation.

When the ISA is subjected to a shock of one standard deviation and the impulse responses to itself, CF and INN are as follows. The response of the CF to the shock of the ISA is negative, reaching a peak in the third period, then slowly decreasing and finally converging to 0. The change in industrial structure leads to a phased change in carbon footprints. The former is the cause, which depends on the inherent law of economic development, while the latter is the result, which is attributed to the positive impact of the ISA on INN decreases after the shock, with a smaller negative impact after period 1.

When the INN is subjected to a shock of one standard deviation, the impulse responses to itself, and CF and ISA are as follows. Among them, when facing the shock from itself, INN shows a strong positive response, and then gradually decreases with the extension of the response period, showing a time-accumulation effect. The CF shows a significant positive effect for the shock of INN and the response peaks in the fourth period after the shock, after which the response degree starts to weaken and gradually level off. This indicates that the “rebound effect” of scientific and technological innovation on carbon footprints as the single measure of scientific and technological innovation adopted by the BTH cannot reduce carbon footprints. The ISA shows a significant positive effect after the shock to INN, which gradually weakens until converging to 0 after period 5, indicating that scientific and technological innovation in BTH has a positive contribution to the industrial structure advancement in the long run. The transformation of scientific and technological innovation helps to realize the transformation of the industrial structure to the green and low-carbon type.

In summary, carbon footprints, industrial structure advancement and scientific and technological innovation in BTH all have certain self-reinforcing effects. The industrial structure advancement has a suppressive effect on carbon footprints, but the effect of scientific and technological innovation on carbon reduction is not obvious. Scientific and technological innovation helps to transform the industrial structure to the advanced level. In addition, the long-term increase in carbon footprints will force cities to accelerate industrial structure adjustment and the output of scientific and technological innovation achievements.

### 5.5. Variance Decomposition Analysis

To further explore the interaction between carbon footprints, industrial structure advancement and scientific and technological innovation, the variance decomposition was used to analyze the strength of the contribution of each structural shock during the change of endogenous variables and to measure the importance of individual variables on the shocks of the remaining endogenous variables [57]. A thirty-period variance decomposition of the variables was performed, and the results are shown in Table 8.

In terms of CF, the contribution of CF to itself reaches 70.9% in period 10. The variance contributions of ISA and INN are 24.3% and 4.8%, respectively. The contribution of CF to its own variance decreases to 68.7% in period 20, and the contribution of ISA increases to 25.4%. The contribution of INN to CF is stable at 5.9%. It indicates that the main variance contribution of CF comes from itself and ISA, and the explanation of CF by ISA gradually increases in the long run.

In terms of ISA, the contribution of CF to ISA is 35.5% in period 10 and decreases to 34.5% in period 20, and it then stabilizes. The contribution of INN to ISA increases to 6.7% in period 30. It can be seen that the explanatory contribution of ISA in the short term is mainly influenced more by CF in addition to itself.

For INN, the variance contribution of itself is 74.1% at period 10, the variance contribution of CF is 1.3%, and the variance contribution of ISA is 24.8%. At period 20, the variance contribution of itself is stable at 73.2%, the variance contribution of CF is stable at 2.0%, and the variance contribution of ISA is stable at 24.8%. This indicates that INN is mainly driven by its own reinforcement, and a small part of the variation comes from the influence of ISA, and the variation in CF has little effect on it.

From the analysis of the above variance decomposition results, we can see that the variance contribution of each variable basically reaches a stable state in the 20th and 30th periods. The variance explanations of carbon footprints, industrial structure advancement and scientific and technological innovation in BTH mainly come from their own strengthening. In the long run, the influence of the industrial structure advancement on carbon footprints is increasing. In the short term, the industrial structure advancement is mainly influenced by carbon footprints in addition to its own reinforcement. The impact of scientific and technological innovation is also increasing. The change in scientific and technological innovation is mainly driven by its own reinforcement, and a small part of the change comes from the influence of industrial structure advancement. Carbon footprints have a smaller impact on it.

## 6. Conclusions

In this study, we include scientific and technological innovation, industrial structure advancement and carbon footprints in the same research framework. Panel data of 13 cities in the BTH from 2006 to 2019 are selected to explore the dynamic relationship among the three variables through the PVAR model. The main findings are presented below.

First, the industrial structure advancement and carbon footprints have a two-way causal relationship with significant influence on each other, and industrial structure advancement explains carbon footprints more strongly. The impact coefficient of industrial structure advancement on carbon footprints with a one-period lag is −0.8766. In the short term, as the process of the industrial structure continues to advance, its effect on carbon footprints also increases. At the same time, the impact coefficient of industrial structure advancement on carbon footprints with one period lag is 0.0671. This indicates that the pressure of the long-term increase in carbon footprints will also accelerate the industrial structure to the advanced level, which is reflected in promoting the transformation of development mode from factor driven to innovation driven. The results of the Granger causality test also further confirm the existence of a two-way causal relationship between the two. In addition, compared with scientific and technological innovation, the industrial structure advancement has a greater impact on carbon footprints, with a variance contribution of 25.4%, which reveals that policy researchers should pay special attention to the emission reduction effect of industrial structure upgrading.

Second, practice shows that there is a “rebound effect” on the impact of scientific and technological innovation on carbon footprints in BTH, but the continuous increase in carbon footprints will force the level of innovation to improve. The carbon footprints tend to increase under the impact of technological innovation, and the coefficient of scientific and technological innovation on carbon footprints with a lag of one period is 0.0992. This confirms the Khazzoom−Brookes hypothesis that the direct green emission reduction effect of innovation is not sufficient to offset the increase in pollutant emissions caused by the expansion of production scale due to the reduction in production cost of enterprises, which eventually results in an increase in carbon footprints. Moreover, with the narrowing of the technology emission reduction space and the gradual speeding up of the carbon market’s construction, the cost of carbon reduction continues to rise, so the single measure of technology innovation cannot effectively achieve the carbon reduction target. In the long term, to achieve effective control of carbon footprints, it is necessary to encourage research and development of green and low-carbon technologies at the practical level. Additionally, the impact coefficient of carbon footprints on scientific and technological innovation with a lag of one period is 0.2120, indicating that the long-term growth of carbon emission will force the government and enterprises to accelerate the R&D of science and technology, especially green innovation, to promote the innovation-driven transformation.

Third, scientific and technological innovation can promote the shift of industries from low-end industries to high-tech industries, thus accelerating the advance of industrial structure. The impact coefficient of scientific and technological innovation on industrial structure advancement in the one lag period is 0.0282. Industrial structure advancement also shows a significant positive effect under the impact of scientific and technological innovation, which gradually weakens and converges to 0 after the fifth period, indicating that scientific and technological innovation in BTH has a positive contribution to the industrial structure in the long run. Meanwhile, the structural contribution of scientific and technological innovation to industrial structure advancement can reach 6.7%. Therefore, we should promote the application and upgrading of green technology to continuously eliminate backward production capacity. While extending the industrial chain and expanding the scale of industry, we can promote the transition of industry to innovation and the deep integration of the innovation chain and industrial chain with the aim of achieving the carbon reduction target and green and high-quality development.

Due to the limitations of the research scope and the availability of data, there are still some limitations in this study. One concern about the findings is with the measurement of variables, especially about the carbon footprints. The widely used methods in studies are the emission factor method, mass balance method and actual measurement method, etc. The results calculated by different measurement methods are slightly different. Another limitation is about the sample size. The external validity of this paper cannot be stated explicitly. Since the BTH is an important urban cluster including the capital of China, the national policy support gives a certain priority to economic development and has a certain reference role. However, this study is still a small sample test. There is heterogeneity in the economic and environmental conditions of different regions, so the external applicability of the study findings needs further analysis. Care should be taken when generalizing the study findings to regions with similar economic characteristics. The impact mechanisms of carbon footprints, such as the impact threshold of variables and regional heterogeneity, should be studied in the future, pending separate analysis.

## 7. Research Implications

The potential contributions of this study are mainly in the following aspects. First, this study uses the IPCC inventory coefficient method to calculate the carbon footprints from direct energy consumption, and sums it with the carbon footprints from indirect energy consumption to finally calculate the total carbon footprints of each city. The carbon footprints of BTH and the change characteristics of scientific and technological innovation are summarized from the temporal and spatial perspectives, which is helpful to visualize its change trend. Second, this study uses GMM estimation for differential equation by one lag order based on the PVAR model to obtain the differential equation. By observing the magnitude and direction of the coefficient values, we explore the static influence relationships between scientific and technological innovation, industrial structure advancement and carbon footprints. Further, through the Granger causality test, we analyze the causal relationships existing between the variables and reveal the bidirectional influence relationships between carbon footprints and their drivers. Finally, by applying Monte Carlo simulation of the response relationships of variables under policy shocks and plotting impulse response plots, we can observe the long-run dynamic impact relationships among variables, which can contribute to the prediction of policy implementation outcomes. In terms of theoretical significance, this study can enrich the PVAR model’s application and research, while confirming the effective applicability of the method in the environmental field. At the practical level, this study takes the perspective of coordinated development of carbon footprints, scientific and technological innovation and industrial structure, and provides targeted ways to promote carbon emission reduction and industrial structure adjustment, which is conducive to providing policy support for the coordinated development of BTH and promoting the realization of the Double Carbon goal.

Based on the above conclusions, we put forward the following policy implications. First, attention should be paid to the inhibiting effect of upgrading the industrial structure on carbon footprints. We should realize that the previous high pollution and rough development mode needs to be changed urgently, and promote the continuous transformation of industry’s structure to the advanced level. The industrial structure, industrial scale and industrial layout should match the energy structure. Second, based on the role of scientific and technological innovation in promoting carbon reduction and industrial structure optimization, we can rely on the fruits of scientific and technological innovation to reasonably optimize the industrial structure and promote the green and low-carbon transformation of industries. At the same time, financial support, tax incentives, property rights protection and other comprehensive policy instruments should be chosen to play the synergistic role of green technology in economic development, so as to jointly achieve the goal of carbon peaking. Third, based on the macro perspective, the mode of energy consumption should be changed by adjusting traditional energy and clean energy demands by introducing pricing mechanisms and expanding clean energy supply to abandon coal-based energy consumption structure. Third, based on the macro perspective, the mechanisms of energy consumption should be changed. Traditional energy and clean energy demands need to be adjusted by introducing pricing mechanisms and extending clean energy’s supply in order to abandon the single energy consumption structure dominated by coal. Meanwhile, promoting the integration of renewable energy with the industrial structure and infrastructure construction should also be particularly noted.

## Figures and Tables

**Figure 1 ijerph-19-09513-f001:**
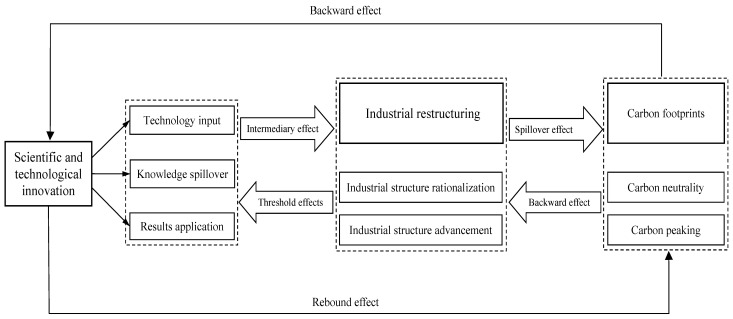
Relationships between scientific and technological innovation, industrial restructuring and carbon footprints. Figure source: authors’ own creation.

**Figure 2 ijerph-19-09513-f002:**
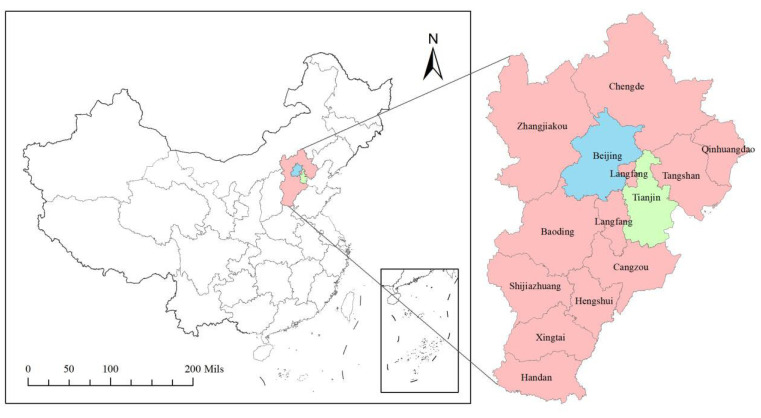
Overview map of the BTH in China. Figure source: authors’ own creation.

**Figure 3 ijerph-19-09513-f003:**
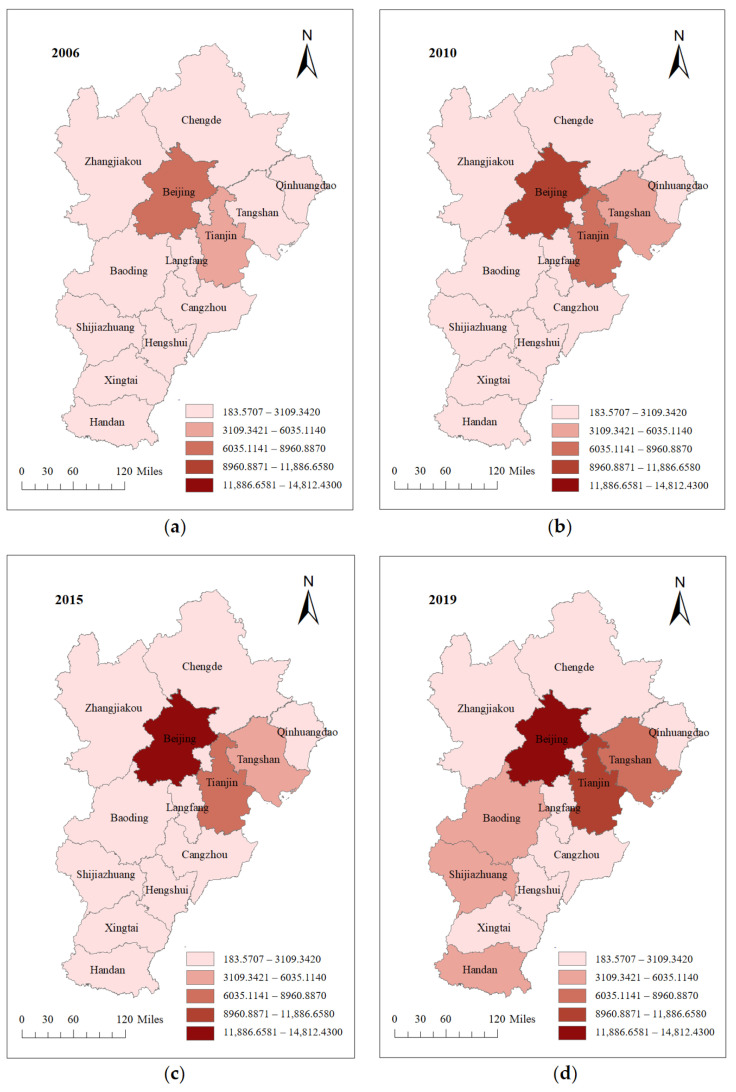
Spatial distribution of carbon footprints in BTH. (**a**) Spatial distribution of carbon footprints in BTH in 2006; (**b**) spatial distribution of carbon footprints in BTH in 2010; (**c**) Spatial distribution of carbon footprints in BTH in 2015; (**d**) spatial distribution of carbon footprints in BTH in 2019. Figure source: authors’ own creation. Data from the *China Energy Statistical Yearbook*, *China Urban Statistical Yearbook* and *2006 IPCC Guidelines for National Greenhouse Gas Inventories*.

**Figure 4 ijerph-19-09513-f004:**
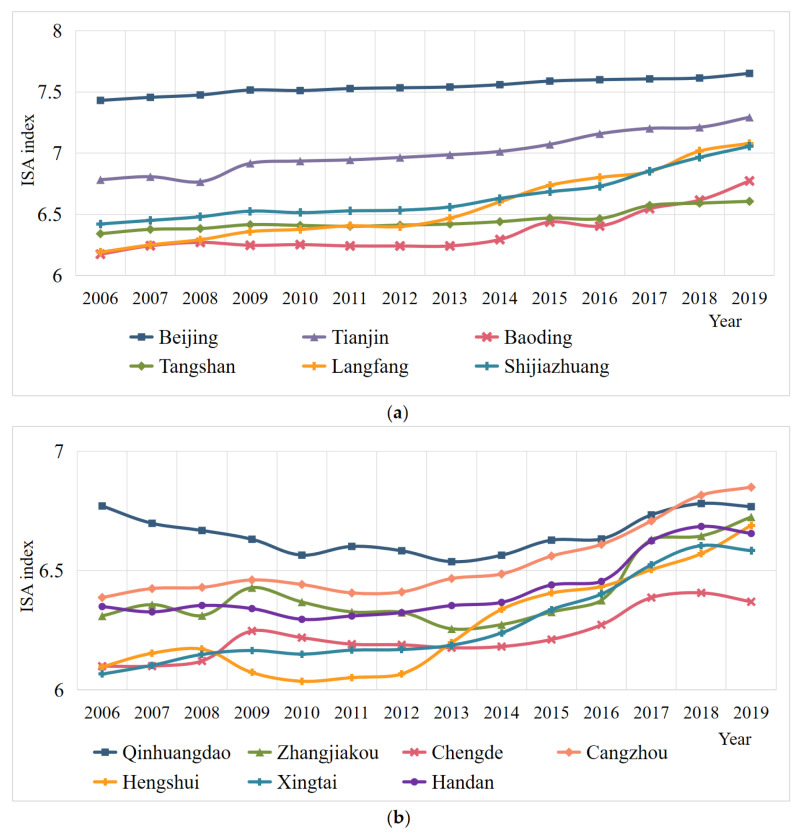
Temporal evolution of industrial structure advancement in BTH. (**a**) Temporal evolution of industrial structure advancement in Beijing, Tianjin, Baoding, Tangshan, Langfang and Shijiazhuang; (**b**) temporal evolution of industrial structure advancement in Qinhuangdao, Zhangjiakou, Chengde, Cangzhou, Hengshui, Xingtai and Handan. Figure source: authors’ own creation. Data from the *China City Statistical Yearbook*.

**Figure 5 ijerph-19-09513-f005:**
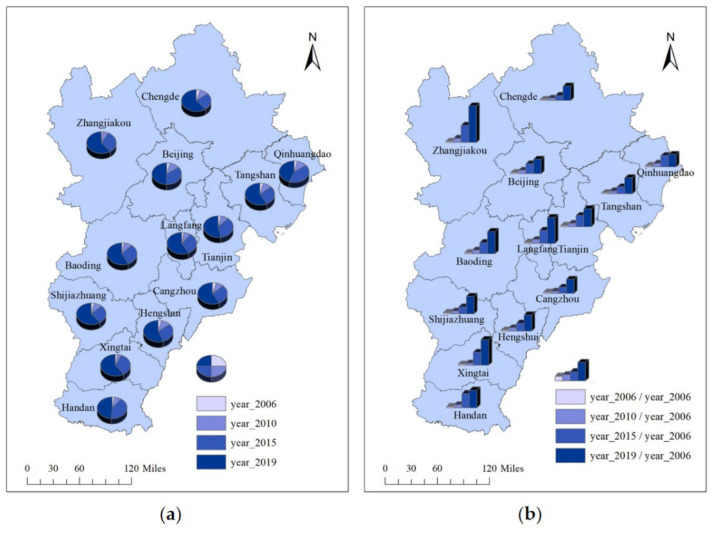
Spatial distribution of patent grants and growth rates in BTH. (**a**) Spatial distribution of patent grants in BTH; (**b**) spatial distribution of growth rates in BTH. Figure source: authors’ own creation. Data from the *China City Statistical Yearbook*.

**Figure 6 ijerph-19-09513-f006:**
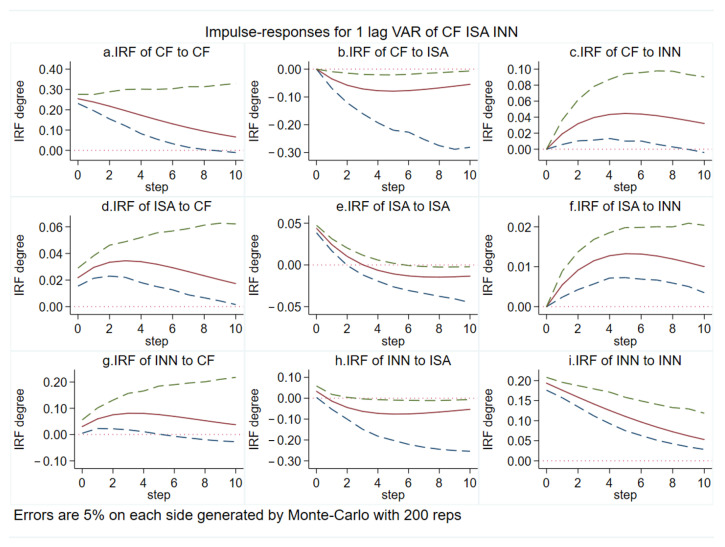
Impulse-response function. Figure source: authors’ own creation.

**Table 1 ijerph-19-09513-t001:** Descriptions of variables.

Variable	Abbr.	Description	Unit
Carbon footprints	CF	The sum of carbon dioxide emissions from fossil energy, transportation, electricity and heat in a city	Million tons
Industrial structure advancement	ISA	The degree of change in the proportion of the three industries rising along the order of primary, secondary and tertiary industries in a city	/
Scientific and technological innovation	INN	The scientific and technological innovation capacity and level of scientific and technological innovation output in a city	Pieces

Table source: authors’ own creation.

**Table 2 ijerph-19-09513-t002:** Descriptive statistics.

Variable		Mean	Std. Dev.	Min	Max	Observations
CF	overall	7.1691	1.1604	5.2126	9.6032	N = 182
	between		1.0683	5.9124	9.3316	n = 13
	within		0.5360	6.0843	8.6912	T = 14
ISA	overall	6.5644	0.3816	6.0366	7.6516	N = 182
	between		0.3598	6.2269	7.5437	n = 13
	within		0.1595	6.1957	7.0848	T = 14
INN	overall	7.4983	1.6156	3.9318	11.7884	N = 182
	between		1.4021	5.7689	10.7499	n = 13
	within		0.8863	5.6255	9.1240	T = 14

Table source: authors’ own creation.

**Table 3 ijerph-19-09513-t003:** Smoothing test.

Variables	Statistics	LLC	IPS	Fisher−ADF	Fisher−PP	Result
CF	z	−3.0563 ***	−1.2563	44.0032 **	50.9189 ***	/
	P	0.0011	0.1045	0.0151	0.0024	
D_CF	z	−10.5343 ***	−5.1794 ***	58.7956 ***	48.1316 ***	Smooth
	P	0.0000	0.0000	0.0002	0.0000	
ISA	z	−2.2438 **	−0.1443	51.8477 ***	78.4992 ***	/
	P	0.0124	0.4426	0.0019	0.0000	
D_ISA	z	−8.1318 ***	−5.6308 ***	91.9612 ***	33.1921 ***	Smooth
	P	0.0000	0.0000	0.0000	0.0000	
INN	z	−2.8042 ***	−3.3297 ***	53.2828 ***	48.7433 ***	Smooth
	P	0.0025	0.0004	0.0012	0.0044	

Note: *** *p* < 0.01, ** *p* < 0.05. D_ indicates first-order difference. Table source: authors’ own creation.

**Table 4 ijerph-19-09513-t004:** Co-integration test.

	Method	z	*p*
Kao	ADF	2.6031 ***	0.0046
Pedroni	Panel ADF	4.0272 ***	0.0000

Note: *** *p* < 0.01. Table source: authors’ own creation.

**Table 5 ijerph-19-09513-t005:** Optimal lag order determination.

Lag	AIC	BIC	HQIC
1	−3.14733 *	−2.20892 *	−2.76619 *
2	−2.74281	−1.56181	−2.26291
3	−2.59617	−1.14034	−2.00462
4	−2.76581	−0.99519	−2.04696
5	−1.05634	1.07951	−0.191047

Note: * denotes the optimal lag order under the three criteria. Table source: authors’ own creation.

**Table 6 ijerph-19-09513-t006:** GMM estimation.

Variables	Statistics	h_CF Equation	h_ISA Equation	h_INN Equation
L1.h_CF	Coef.	0.9987 ***	0.0671 ***	0.2120 **
	z	9.09	2.76	2.32
L1.h_ISA	Coef.	−0.8766 **	0.5334 ***	−1.0002 **
	z	−2.03	5.17	−2.24
L1.h_INN	Coef.	0.0992 *	0.0282 ***	0.9089 ***
	z	1.90	2.90	19.22

Note: *** *p* < 0.01, ** *p* < 0.05, * *p* < 0.1. D_ indicates first-order difference. Table source: authors’ own creation.

**Table 7 ijerph-19-09513-t007:** Granger causality test.

H_0_: the Former is Not the Granger Reason for the Latter	Chi-Square	*p*	Result
h_ISA→h_CF	4.1319 **	0.042	reject
h_INN→h_CF	3.6234 *	0.057	reject
all→h_CF	4.7662 *	0.092	reject
h_CF→h_ISA	7.6131 ***	0.006	reject
h_INN→h_ISA	8.3988 ***	0.004	reject
all→h_ISA	13.517 ***	0.001	reject
h_CF→h_INN	5.3908 **	0.020	reject
h_ISA→h_INN	5.0251 **	0.025	reject
all→h_INN	5.6524 *	0.059	reject

Note: *** *p* < 0.01, ** *p* < 0.05, * *p* < 0.1. Table source: authors’ own creation.

**Table 8 ijerph-19-09513-t008:** Variance decomposition.

Variables	s/Period	CF	ISA	INN
CF	10	0.709	0.243	0.048
	20	0.687	0.254	0.059
	30	0.687	0.254	0.059
ISA	10	0.355	0.590	0.055
	20	0.345	0.589	0.066
	30	0.345	0.588	0.067
INN	10	0.013	0.246	0.741
	20	0.020	0.248	0.732
	30	0.020	0.248	0.732

Table source: authors’ own creation.

## Data Availability

The datasets used or analyzed during the current study are available from the corresponding author on reasonable request.

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
