# Peer review of "The Impact Relationships between Scientific and Technological Innovation, Industrial Structure Advancement and Carbon Footprints in China Based on the PVAR Model"

_ijerph, 2022, doi:10.3390/ijerph19159513_

Round 1
Reviewer 1 Report
The topics discussed in the manuscript are interesting, and author(s) provided sufficient background and included all relevant references in Introduction section.
The research design is appropriate.
The method and analysis sections adequately have been described.
The results have been clearly presented.
The conclusions have been strongly supported by the results.
I don't feel qualified to judge about the English language and style.
Generally, I believe that the front-end of the paper is well-written, but I think it could benefit with a bit more clarity.
However, my comments are as follows:
Better explain the urgency and importance of its investigation from global point of view, clearly identify how will our readers benefit from the investigation of this research.
#Line 172-174 Previous studies have not been conducted only at the national scale, but also at the regional or city scale, whether in the Chinese or English literature.
#Line 175 "ignoring the endogenous variables." This is a crucial point made by the author, but it is not stated with enough specificity. Which specific endogenous variables are being ignored.
The abbreviations of the variables in Table 1 were not mentioned in the previous section. Instead, they are placed in 3.2 Variable Descriptions. This presentation is rather unfriendly to the reader.
#Line 301 Due to the inconsistent range of values for the legend colour blocks, these four charts cannot be compared in one piece. It is recommended that the range of values for the different colour blocks in the legend be standardised.
#Line 318-330. In my opinion, this section does not accurately characterize spatio-temporal evolution. It merely shows the temporal variation of the relevant indicators on the map.
Propose some improvements and direction for future research.
Reviewer 2 Report
To investigate the dynamic relationships between scientific and technological innovation, industrial restructure advancement and carbon footprints, this paper selected panel data of the three variables in BTH from 2006-2019, and established a Panel Vector Auto-Regressive (PVAR)model to conduct an empirical study by means of systematic Gaussian Mixture Model (GMM) estimation, impulse response analysis, and variance decomposition. However, there are two main issues that could be improved.
1. In the introduction, the reasons for this study are still not sufficiently explained.
2. By using various methods, this paper has produced some results, but the discussion of these empirical results is still inadequate.
Reviewer 3 Report
Thanks for giving me the opportunity to evaluate this paper
The idea of the paper is interesting and in my opinion, this paper can be accepted for publishing after some minor corrections
I suggest some improvements aimed to increase the quality of the paper:
1).The paper clearly lacks the sense of paragraphing; some paragraphs are too shorts and consist of only few lines. Kindly rearrange your paragraphs and provide arguments about one main theme in each paragraph.
2. Recheck throughout the paper and ensure that all abbreviations are defined the first time they are used.
3). Methods seem fine to me.
4). The Results and discussion described with reference to the references of the parameters that influence the results of the research will make it more original. The purpose of the study and the reasons clearly set forth.
5).Improve the policy suggestions
6). Overall, the quality of English used in this study requires significant improvement.
Reviewer 4 Report
The authors submitted a manuscript on a current environmental topic. Specifically, the authors in the research deal with the carbon footprint and greenhouse gas emissions.
In selected industrial Chinese regions, they use statistical methods to examine numerical series of data in an attempt to assess the interaction of Scientific and Technological Innovation, Industrial Restructure Advancement and Carbon Footprints in China based on PVAR Model.
At the beginning of my review, I will list the strengths and weaknesses of the publication, and later I will qualify specific caveats that need to be addressed.
I perceive the topicality of the topic, the analysis of the impact of innovations and green technologies and their subsequent effect on the carbon footprint, and the analysis as a whole as strengths of this publication.
The weak point is the poor presentation of the results achieved and the construction of the article in general, where basic errors in all sections can be qualified.
Specific defects:
- The abstract section lacks a brief summary of the achieved results in numbers. Here you need to be brief but specific and state the results achieved in numbers, percentages, tons, etc.
- The Introduction section lacks a precise specification of the main and secondary objectives of the research.
In the introduction, it is also necessary to explain why individual cities were chosen for analysis and their specifics, and here or in the Methodology section, individual variables should be better described.
- Sources must be cited for all figures, graphs and tables. Even though graphs and tables can be assumed to be the authors' own work and presentation, it is still necessary to state the Source: Own creation, etc. For example, in the picture of the interrelationships of the investigated phenomena, the source is not apparent and it may be a borrowed picture.
- On line 182, at the end of the page, there is the title of the Methodology and Data section itself, which must be moved to the next page, and from the point of view of presentation, the mere title of the chapter and the following sub-title of the next chapter are unacceptable. At a minimum, the title of the chapter should state: "This chapter will deal with..."
- Another major drawback of the presentation are formulas, see lines 244, 246, 267, which do not have individual elements subsequently described. After specifying the formula, it is necessary to specify WHERE: and describe the individual variables...
- Charts, e.g. lines 320, 321, 479 do not have described axes.
- The Discussion section is partially replaced in the Conclusion section, however, I lack a comparison with other scientific works on this topic. The topic is covered abundantly and well in scientific articles.
- The Conclusion section absolutely lacks an evaluation of the established research objectives and unfortunately there is no precise quantification of the achieved results of the analysis, again in specific units. A general summary of the results is so inadequate!
Here I must also contradict the erroneous use of the term NEGATIVE EXTERNALITY, where the authors present it as a consequence of innovation. It should be noted that a negative externality is extensive production, when the producer produces greenhouse gases and generates profits.
- Chapter 6.2 contains recommendations that are very general and flat. It should be noted that innovation and green technologies act synergistically in the mix of other state policy instruments, such as Institutional, legal, economic, etc. instruments.
Reviewer 5 Report
Although the article presents research on only one country, its high level of content should be appreciated. The authors have presented their argument in a logical and coherent manner. Nevertheless, the following changes are recommended: (1) in the introduction, in addition to the main objective, the specific objectives and research hypotheses should be presented; the authors could also describe the structure of the article; (2) table 1 could be moved to subsection 3.2.; (3) the recommendations that are found in the last section should be moved to the section with the research results; (4) the conclusion lacks the limitations of the conducted research and future research directions; (5) the authors should also highlight the contribution of the article to the development of theory and practice.
Round 2
Reviewer 4 Report
The study is improved.